# Spatial Association of Urbanization in the Yangtze River Delta, China

**DOI:** 10.3390/ijerph17197276

**Published:** 2020-10-05

**Authors:** Wei Zhao, Xuan Liu, Qingxin Deng, Dongyang Li, Jianing Xu, Mengdi Li, Yaoping Cui

**Affiliations:** 1Key Laboratory of Geospatial Technology for the Middle and Lower Yellow River Regions, Henan University, Ministry of Education, Kaifeng 475004, Henan, China; 10130056@vip.henu.edu.cn (W.Z.); lmd@cug.edu.cn (M.L.); 2College of Environment and Planning, Henan University, Kaifeng 475004, Henan, China; 1610131043@vip.henu.edu.cn (X.L.); 104753190125@henu.edu.cn (D.L.); 3School of Urban Planning and Design, Peking University, Shenzhen 518055, Guangdong, China; dengqx@pku.edu.cn; 4College of Land Science and Technology, China Agricultural University, Beijing 100193, China; 1610130055@vip.henu.edu.cn

**Keywords:** spatial polarization, spatial diffusion, urban evolution, urbanization, internal mechanism

## Abstract

China is urbanizing rapidly, but current research into the spatiotemporal characteristics of urbanization often ignores the spatial and evolutionary associations of cities. Using the theory of spatial polarization and diffusion, together with a systematic analysis method, this study examined the spatial development process of urbanization in the Yangtze River Delta (YRD) region of China during 1995–2015. Results showed clear patterns in the scale and hierarchy of regional urbanization. Shanghai ranked first as the regional growth pole, while Nanjing, Hangzhou, and Suzhou ranked second. The spatial linkage index of urbanization showed that 10 cities (including Shanghai, Suzhou, and Hangzhou) constituted the densest spatial linkage network. The diffused area often became spatially polarized before the polarization then weakened as a new diffusion stage developed. The study also revealed that the spatial correlation urbanization differences in the YRD generally decreased. The polarization index revealed increasing spatial integration and correlation of urbanization in the YRD. This study proved that each city had a different spatial role in relation to other cities during different stages of development. Investigation of the driving mechanism of regional urbanization indicated that industrial modernization and relocation within the region provided the main endogenous driving force for the formation of spatial polarization or diffusion. Our research provides important scientific support for regional development planning. Furthermore, our analysis of the impact of spatial correlation within cities or a region could provide an important reference in relation to the regional environment and public health.

## 1. Introduction

Uncontrolled urban expansion and development causes considerable ecological and environmental pressure. China has urbanized rapidly in recent years, and it is now in a critical period with regard to the transformation and development of its urbanization process [1]. 

The regional differentiation of China’s urbanization process has received considerable attention. Studying the spatial relationships of regional urban development and the laws of urban spatial polarization and diffusion can provide practical guidance for the planning of regional cities and agglomerations.

In the process of urbanization, cities within a region tend to exhibit a spatial pattern of polarization and diffusion. Spatial polarization refers to the process of spatial differentiation and agglomeration of various economic and social factors, and the formation of growth centers within a certain geographical area through resource allocation. This dynamic process is a vital driving force for regional development [2]. Generally, the “pull” or primacy of a city reflects the different degrees of polarization within a region. In contrast, spatial diffusion refers to the process of regional coordination or simultaneous development through spatial interaction within a region. Spatial polarization and spatial diffusion are two distinct phenomena of the urbanization process that interact to promote or constrain regional urban development.

As the spatial correlation aspects of urbanization are obvious, the phenomenon of spatial polarization and diffusion has attracted interest globally. Urban polarization is found in both developed countries (e.g., the United States and Germany) and developing countries (e.g., Russia) [3,4,5]. Research suggests that the world economy became more polarized between 1960 and 1999, and that access to the World Trade Organization did not reduce differences and levels of polarization among member states [6]. Others argue that levels of urbanization show significant spatial dependence [7,8]. It has also been reported that highly concentrated urbanization might hinder sustainable regional development [9]. Many international studies have conducted urban network research based on urban networks and mobile space theory. However, spatial linkages and interactions between regional cities and urbanization processes cannot be measured simply through urban networks [10,11].

The characteristics of urbanization and its evolution can be reflected in changes of the regional industrial structure, and the speed of regional urbanization influences the spatial transfer of industries [12,13]. In China, there are significant regional differences at the provincial scale between the levels of urbanization and economic development, and regional polarization is evident [14,15,16]. The continuous expansion of urban agglomerations and urban bay areas also highlights the guiding role of regional urbanization in regional development [17,18]. Research into this by Chinese scholars has manifested in two main areas. The first area concerns research on the impact of the urban networks proposed by international studies from multiple scales (e.g., regional cities, urban agglomerations, monolithic cities, and inner-city space) to multiple fields (e.g., population, capital, transportation, and science and technology flows). Most of this research involved analysis of urban spatial correlation from the perspective of urban networks, central cities, and spheres of influence [11,19,20,21,22]. Although network research can indicate the mobility of various urbanization elements through a vector of points and lines, it is difficult to express the urbanization of a region as a surface. The second area concerns research on the role of urban space in relation to a single indicator such as population. However, few studies have systematically analyzed and measured the spatial correlation characteristics of regional cities, combining spatial polarization and diffusion by integrating multiple indicators such as population, economy, and land use under the dual dimensions of time and space. Regional cities are closely interconnected and spatially linked, not only in the spatial polarization of single urbanization elements, but also in the development of spatially linked features over time [23].

Expansion of cities and the flow of various elements among urban developments are vital to the spatial correlation of urbanization. The spatial correlation of urbanization focuses on revealing the characteristics of regional interaction. It is also an interactive process of various elements in urban development, which refers to population mobility and city networks [11]. Population, resources, elements, and industries tend to gather into developed areas on a large scale to form several spatially distributed densely populated urban areas [24]. Urbanization change can be measured by establishing a “population–land–industry” indicator system to capture its spatiotemporal variations [25]. Previous studies have revealed the internal correlation of urban development within China using comprehensive quantitative methods, e.g., the spatial econometric regression model, entropy method, Moore’s structural change index, and Moran’s *I* index [26]. Most studies tend to measure the level of urbanization by calculating the Moran’s *I* index [27,28,29]; however, the Tsui–Wang (TW) index has been applied to analyze the overall polarization trend in related research [30].

The objective of this study was to elucidate the spatial association of cities and the spatial development process within the Yangtze River Delta region. The YRD is China’s largest economic core region and one of the regions in China with the highest level of urbanization. The region has also become a focus for study of the modern urbanization process [31,32,33,34]. Therefore, analyzing the spatial and evolutionary associations of cities within the region from different perspectives is vital with regard to research on the urbanization process in the YRD. This study used economic, demographic, and land use data to construct urbanization spatial development indices with which to measure both the scale of development and the strength of the spatial linkages between cities under regional urbanization. Spatial autocorrelation analysis and a polarization index were used to quantify the differences in spatial development within the YRD. Finally, systematic analysis of the spatial polarization or diffusion of urban development was performed by integrating a spatial growth index of urbanization. As the largest area of globalization in China, the YRD has become synonymous with population, economic, and urbanization development. The findings of this study could represent an important resource for the future healthy development of cities in China.

## 2. Materials and Methods

### 2.1. Study Area

The YRD region, which covers 350,000 km^2^, has the most developed economy, the densest population, and one of the highest levels of urbanization in China [35,36]. In recent decades, the YRD has experienced rapid urbanization [37]. Exploring the pattern, process, and internal mechanism of the urbanization process within the YRD is of great importance for maintaining regional security and promoting sustainable development. The region includes 41 prefecture-level cities in Shanghai, Jiangsu, Zhejiang, and Anhui, all of which are located on the middle and lower stretches of the Yangtze River plain (Figure 1). Constraints on production activities attributable to regional topographic and environmental differences have affected the distribution of the population, i.e., there are more people in the northern parts than in the south. According to data from the National Bureau of Statistics in 2015, the GDP of the YRD region exceeded 16 trillion yuan, which accounted for almost 1/4 of the national total. The average urbanization rate of China was 56.1%, whereas the urbanization rates of Jiangsu, Zhejiang, Anhui, and Shanghai were 67.7%, 67.0%, 51.0%, and 87.6%, respectively.

### 2.2. Data Sources

In this study, data of city-level GDP, registered household population, and the urban built-up area of the YRD were used as indicators with which to measure regional economic urbanization, population urbanization, and spatial urbanization, respectively. The data were obtained from the Shanghai, Zhejiang, Anhui, and China City Statistical Yearbooks and the Yangtze River Delta Science Data Center of the National Earth System Science Data Center (http://nnu.geodata.cn:8008). Four periods were considered from 1995 to 2015, each covering a five-year interval. As there were missing data for some cities that were affected by administrative zoning adjustments or line changes during the study period, this study split the cities into their smallest administrative units and merged the data between 1995 and 2015 from the smallest administrative units into the newest subdivision of the city limits in 2015.

Urban spatial correlations were measured by integrated distance. Straight-line distance, highway distance, and high-speed rail distance were three specific indicators selected with reference to the first law of geography [11,22]. Straight-line distance was calculated from the latitude and longitude coordinates of two government sites. The highway distance between the two places was determined as the length of the shortest route on the Baidu map. The high-speed rail distance between the two places was obtained from the China Railway Corporation.

### 2.3. Urban Spatial Association and Evolution

#### 2.3.1. UD and UCI

We used the metrics of overall urbanization development (UD) and urbanization spatial correlation intensity (UCI) to reflect the relative overall power of a city within the study area, changes in development within the region, and the strength of the spatial linkages in urban development. Urban economy, urban population, and urban space were linked to determine the scale and spatial distance correlation of cities based on the first law of geography and relevant literature [11,22,38,39]:
(1)
$$d_{i}=\sqrt[{3}]{p_{i}{\times l}_{i}{\times e}_{i}},$$


(2)
$$R_{ij}{=\varphi }_{1}r_{zx}{+\varphi }_{2}r_{gl}{+\varphi }_{3}r_{gt},$$


(3)
$${UD}_{i}=\frac{d_{i}}{\sum_{i=1}^{n}d_{i}},$$


(4)
$${UCI}_{ij}=\frac{d_{i}\times d_{j}}{R_{ij}{}^{2}},$$

where 
$p_{i},\;l_{i}$
, and 
$e_{i}$
 represent the registered household population, the built-up area, and the GDP of city *i*, respectively; 
$\varphi_{1}$
, 
$\varphi_{2}$
, and 
$\varphi_{3}$
 are the weighted indices; 
$r_{zx}$
, 
$r_{gl}$
, and 
$r_{gt}$
 represent the straight-line distance, highway distance, and high-speed rail distance, respectively; 
$R_{ij}$
 is the integrated distance between city *i* and city *j*. The larger the value of 
$UD_{i}$
, the stronger the integrated urbanization of the city. A larger 
$UCI_{ij}$
 indicates stronger spatial distance correlation between city *i* and city *j*.

#### 2.3.2. Urban Spatial Differentiation and Relations

In this study, the Moran’s *I*, local indicators of spatial association (LISA) agglomeration, and the polarization index were used to measure the differences in spatial development level and spatial relationship in the YRD region. Local autocorrelation analysis and the polarization index were complementary and measured changes in spatial agglomeration or dispersion that indicated different levels of urban development. Moran’s *I* and LISA agglomeration are commonly used indicators of spatial autocorrelation [27,28,29]. The polarization index quantified the degree of polarization of regional development, but it could not indicate the specific regions of polarization in space. Based on the LISA agglomeration maps, we could observe the diffusion effects and polarization characteristics of the region.

(1) Spatial heterogeneity

A simple binary adjacency matrix was used to represent the spatial proximity relationship of *n* positions and to determine the spatial weight matrix. We defined the value of the spatial weight matrix as 1 (0) when region *i* was adjacent (not adjacent) to region *j*. The local Moran’s *I* was used to detect spatially differentiated features. The local Moran’s *I_i_* can be expressed as follows:
(5)
$$I_{i}{=Z}^{\prime }\sum_{j}w_{ij}Z_{j}^{\prime },$$

where the standardized statistics for the *I_i_* test can be calculated using the following formula:
(6)
$$Z\left(I_{i}\right)=\frac{I_{i}{‒E(I}_{i})}{\sqrt{{VAR(I}_{i})}},$$

where *Z′_I_* and *Z′_j_* are standardized observations, and *w_ij_* is the matrix of spatial weights that indicates the proximity of city *i* to city *j* in the region. A positive value of *I_i_* indicates spatial agglomeration of similar values (high or low) around the regional unit, and a negative value of *I_i_* indicates spatial agglomeration of non-similar values. When *Z*(*I_i_*) is positive and significant, there is positive spatial correlation within the region; there is negative spatial autocorrelation when *Z*(*I_i_*) is negative and significant, and when *Z*(*I_i_*) is 0, the observations are distributed randomly [39].

(2) Spatial polarization level

The TW index is based on the Wolfson index and it is summarized by the two partial ordering axioms of increasing polarization and diffusion. It can be used to describe the degree of spatial polarization; the larger the value, the greater the regional spatial polarization [30]. The TW index can be expressed as follows:
(7)
$$\mathrm{TW}=\frac{\theta }{N}\sum_{i=1}^{k}\pi {\left|\frac{y_{i}-m}{m}\right|}^{r},$$

where 
N
 is the total population within the region, 
θ
 is the population of geographic area *i*, *k* is the number of geographic areas, 
$\pi_{i}$
 is the population of geographic region *i*, 
$y_{i}$
 is the real GDP per capita for geographic region *i*, *m* is the median of real GDP per capita for the entire geographic region *i*, and *r* lies between 0 and 1. In this study, we took 
θ
 as 1 and *r* as 0.5. The range of TW is from 0 (no polarization) to 1 (full polarization).

#### 2.3.3. Spatial Evolution

Factors such as population, built-up area, and urban industry all have significant impact on the process of urbanization. The urbanization speed (US) was determined from the differences in the relative development rates among cities. US reflects the overall development of a city:
(8)
$${US=\alpha }_{1}P_{i}{+\alpha }_{2}L_{i}{+\alpha }_{3}E_{i},$$

where 
$\alpha_{1}$
, 
$\alpha_{2}$
, and 
$\alpha_{3}$
 are index weightings; 
$P_{i}$
 is the speed of population development; 
$L_{i}$
 is the speed of urban land use change, and 
$E_{i}$
 is the speed of urban economic development. The speed of population development 
$P_{i}$
 can be expressed by the following logistic equation:
(9)
$$P_{i}=\frac{d_{P}}{d_{t}}=rP_{0}\left(1-\frac{P}{K}\right),$$

where *P* is the population within the region, 
$P_{0}$
 represents the initial population size, *r* represents the maximum relative rate of regional growth that can be driven given the limiting factors in regional population growth, and *K* denotes the highest regional population that can be driven given the limiting factor in regional population growth where *P_max_* = *K*. The speed of urban land scale 
$L_{i}$
 and the speed of urban economic development 
$E_{i}$
 can be expressed as follows:
(10)
$$L_{i}=\sqrt[{n}]{\frac{q_{n}}{q_{0}}}-1,$$


(11)
$$E_{i}=\sqrt[{n}]{\frac{e_{n}}{e_{0}}}-1,$$

where 
$q_{n}$
 and 
$q_{0}$
 are the urban built-up area at the end of the study period and the initial year, respectively, 
$e_{n}$
 and 
$e_{0}$
 are the urban GDP at the end of the study period and the initial year, respectively, and *n* represents the study period.

## 3. Results

### 3.1. Urban Spatial Correlation Characteristics

The UD and UCI reflect the relative overall importance of cities and the variation within the region in space and time. This study used natural breaks to classify UD and UCI [40]. Overall, the pattern of importance within the YRD region did not change obviously. The UD in Shanghai decreased gradually from 0.15 to 0.12 during the study period. Similarly, Jiangsu Province showed a downward trend in UD from 0.38 to 0.36 in the first five years, but it subsequently increased to 0.41. However, the UD in both Zhejiang and Anhui provinces remained stable with no obvious change between 1995 and 2005 (Figure 2). During 1995–2015, the hierarchical structure of urban development showed five tiers in terms of spatial distribution (Figure 3). Shanghai was in the first tier, and Suzhou was added to the second tier alongside Nanjing and Hangzhou. The third tier was dominated by Ningbo, Wuxi, Wenzhou, Nantong, Hefei, and Xuzhou, and the remaining cities were in the fourth or fifth tiers.

The UD ranking over five periods revealed that the cities that were always in the top 10 were the major cities that constituted the region’s urban centers (Table 1). Jiangsu, Zhejiang, and Anhui provinces accounted for 50%, 30%, and 10% of the cities, respectively. The UD values for these 10 cities varied over time, but this was mainly reflected in changes of their ranking (Table 1 and Figure 3). Shanghai maintained the number 1 rank in the region, but its UD value decreased from 0.147 to 0.124 over the 20-year period. The rankings of Nanjing, Hangzhou, and Suzhou also remained stable at second–fourth, respectively, although their UD values increased. Of the remaining cities, Wenzhou and Hefei changed the most during the study period; Hefei moved up from tenth (0.029) to fifth (0.042), while Wenzhou dropped from fifth in 2000 (0.041) to tenth (0.034) in 2015. This variation was a consequence of the changing dynamics and rates of urbanization.

The distance correlation strength of urbanization in the YRD region also showed obvious spatial distribution characteristics (Figure 4), and the high UCI grades were centered in the east. Shanghai, Suzhou, Wuxi, and Changzhou were strong links in the development of urbanization, and the urbanization development of Suzhou and Wuxi showed a “twin star” structure with strong spatial links. The connections among these cities were in the first and second grades. Nanjing and Hangzhou formed the cores of their own urbanization network. Hangzhou and Shaoxing had the strongest correlation followed by Shanghai and Jiaxing, while Nanjing had strong spatial links with Yangzhou and Zhenjiang. Overall, Shanghai, Suzhou, Nanjing, and Hangzhou constituted the most densely urbanized spatial network, which indicated that there were obvious spatial agglomerations and connections in the development of the YRD region. The UCI showed the strength of the spatial linkages of each city and it indicated their respective impact. From this perspective, the UCI to some extent represented the network of interconnection.

### 3.2. Urbanization Spatial Characteristics and Development

#### 3.2.1. Spatial Differentiation and Hierarchical Characteristics of Urbanization

This study used the local Moran’s *I* to derive Moran’s *I* scatter plots and LISA plots for the five equal periods from 1995 to 2015 in the YRD region. The spatially significant positive correlation of GDP per capita in the YRD region in each year indicated that economic development in this region was spatially highly agglomerated (Figure 5). Regions around areas of high GDP per capita also had high values. However, this feature weakened over time. During the study period, the Low–Low (L-L) zones within the YRD region reduced from nine to six, while the High–High (H-H) zones increased from six to seven (Figure 6).

From Figure 6, it can be seen that the northwestern region of the YRD was an L-L cluster, and that the GDP per capita of these cities was relatively low in comparison with the entire YRD region, which showed low correlation. The H-H regions were distributed mainly in the cities surrounding Shanghai, and these cities were highly correlated. The H-L regions that were distributed mainly around the L-L region were significantly polarized. Most of the H-L regions were converted from the former L-L region. The L-H regions were the sedimentation region; they were mainly located around the H-H regions and there was conversion of L-H regions into H-H regions (Figure 6). The L-L zones decreased over the 20-year period and Hefei became the main growth city of Anhui Province, showing transformation from an L-L zone into an H-L zone. However, the H-H zones with Shanghai at the center continued to increase and spread into surrounding areas. For example, Taizhou changed from an L-H zone into an insignificant zone, before finally becoming an H-H zone, indicating that it was influenced by radiation of H-H urban agglomerations.

Generally, the number of cities with mid-range economic development in the YRD region increased rapidly. From the significance level of the Moran’s *I* change and local autocorrelation, the L-L zones as a proportion of the total zones dropped from 52.9% to 40.0%; however, the proportion of H-H zones increased from 35.3% to 46.7%. This indicates that spatial polarization between cities in the YRD region decreased and developed toward spatial balance. The Moran’s *I* decreased from 0.54 to 0.45, which also reflected the trend of diminishing differences in urbanization in the YRD region in the overall space.

#### 3.2.2. Spatial Polarization and Diffusion

The polarization level of Jiangsu Province showed a variable trend according to the TW index of each province in the YRD region. The TW index in Anhui Province grew significantly from 0.45 to 0.65 over the 20-year period. The TW index for Zhejiang Province, with the lowest level of polarization among the three provinces, remained stable at around 0.5 between 1995 and 2003, and then increased to 0.6 in 2015. It is noteworthy that the gap in the polarization index gradually narrowed for all three provinces after 2005 (Figure 7).

Overall, polarization within the region was pronounced. The polarization index of the YRD region reached its highest value around 2003 and then declined. It remained stable at around 0.75 after 2010, which indicated increasing spatial integration and connection of urbanization in the YRD region. This was because of the cascading economic development from coastal/river cities to inland areas. Shanghai was the first city in the region to benefit from this, and then some of its production began to move to neighboring cities. Suzhou and Wuxi were the first to receive industrial radiation from Shanghai because of their strong spatial connections with the city. In recent years, the main growth poles of the YRD region have evolved from points to bands and extended inland, indicating that the radiation effect of the core growth area has reached inland areas. Comparison of the TW index and Moran’s *I* in the YRD region revealed that both indices first increased and then decreased (Figure 8). Polarization in the YRD region intensified between 1995 and 2003, but then weakened as spatial development subsequently became more balanced.

### 3.3. Spatial Correlation Characteristics of Urbanization

The rate of development in these cities was exemplified by the spatial correlation of urbanization. The urbanization of the YRD region showed characteristics of spatial diffusion. Spatially, the high-value area shifted from the coast to the interior, and it showed a changing trend from east to west and south to north. The high-value area of US from 1995 to 2000 was distributed mainly in northern Jiangsu, Hangzhou, Wenzhou, and Shanghai, while the low-value area was located mainly in south-central Anhui and central Jiangsu. However, after 2010, the high-value area of US was concentrated in Anhui and central and northern Jiangsu, while the low-value area was distributed mostly in areas south of the Yangtze River (Figure 9).

There were distinct phases in the rate of urbanization specific to each city. Hefei had a remarkable polarization effect in Anhui Province as its US increased significantly between 1995 and 2010, and the value remained higher than that of its neighbors after 2010. It was not possible to observe the progress of Shanghai’s polarization enhancement to the periphery within the study period, but it was clear that Shanghai’s pattern of diffusion spread inland after 2000. Hangzhou, Nanjing, and Suzhou were all the focus of increasing polarization, which then weakened in the subsequent process of urbanization (Figure 9).

Shanghai, Nanjing, Hangzhou, Suzhou, and Hefei were the five core cities in the YRD region. There were significant differences in their levels of urbanization, but these differences declined over time. The rates of economic growth of Shanghai, Hangzhou, and Suzhou were the most rapid between 1995 and 2000, after which Hefei and Nanjing began to catch up. In terms of population, Hefei, Hangzhou, Nanjing, and Suzhou had the rates of highest population growth. The growth rate of the built-up area in Hangzhou and Shanghai ranked first between 1995 and 2000, while that of Shanghai, Suzhou, Nanjing, and Hefei ranked first after 2000. Hangzhou, Suzhou, and Hefei experienced faster overall urbanization from 1995 to 2005, but it slowed after 2005. Conversely, Nanjing and Hefei maintained relatively high rates of growth (Figure 10). In the overall process of urbanization, economic urbanization was preemptive, land urbanization was generally consistent with economic growth, and population urbanization had a certain lag. Furthermore, the urbanization process was spatially diffuse and recursive. Spatially, cities with higher levels of urbanization drove the development of neighboring cities.

## 4. Discussion

The spatial polarization or diffusion of urbanization in the YRD region has been related to economic activities, especially industrial modernization and relocation. Based on industrial location theory, industrial location is influenced by multiple factors such as transportation, the labor force, and markets. Enterprises will constantly adjust their production and operation activities according to their own positioning and profitability, and will choose the most suitable production area and market place [41]. Different industries and firms will respond differently to each influencing factor and will exhibit different relocation sequences [42,43,44]. Studies revealed that urban factor mobility and industrial relocation in the YRD region have led to changes in the spatial and industrial structure. For example, the industrial differences between Shanghai and Zhejiang promoted the spatial relocation of manufacturing-centered industries in the YRD region [45]. Our study also showed that the effects of industrial specialization are geographically concentrated within the YRD [34]. In addition, there was a significant spatial effect of industrial land prices in different cities on the scale of industrial diffusion. Areas with higher prices for industrial land led to the establishment of efficient enterprises, while inefficient enterprises moved away [46]. The theoretical model of industrial transfer also showed large-scale transfer of China’s manufacturing industry from the east to central and western regions, following the inverse order of the elasticity of industrial substitution. In the study area, this manifested where low substitution elastic industries showed hierarchical diffusion and high substitution elastic industries showed diffusion [47].

Industrial modernization drove the hierarchy and differentiation of industries in regional space, which in turn drove the spatial shift of industries. To some degree, the spatial change of the regional industrial structure reflected industrial relocation, and it revealed the characteristics of urbanization and the underlying reasons for its evolution. In fact, industrial mobility was profoundly reflected in the process of regional economic development, and the phased changes in the spatial growth index reflected this dynamic (Figure 9). The process will push low-end industries out in the course of modernizing industrial structure. Cities with location advantages are the first to relocate and promote regional development in the spatial pattern of industrial transfer [12,47]. This is then manifested in economic, population, and land urbanization [13]. Spatial verification of industrial relocation can also be confirmed through changes in the US of a city (Figure 10). As Shanghai completed its industrial modernization and optimization, low-end industries relocated which prompted a spatial change in the industrial structure [48].

In addition, the urban spatial development pattern and evolution in the YRD region has been affected by national macro policy. The establishment of coastal open cities has improved the regional economic system and facilitated rapid urbanization [49]. Under the policy that accelerated the opening of inland regions to the outside world, the YRD region has introduced multiple development opportunities in coastal and river areas. This policy was bound to exacerbate differences in regional development over time [50]. The coexistence of urban spatial polarization and diffusion will lead to the overall development of a region.

Urbanization factors such as the population and the economy are inseparable from urban development. The environment and public health not only relate to urban planning and development, but also are indispensable with regard to urban spatial association. Accelerated urbanization will lead to spatial polarization and diffusion that could have an impact on the environment and public health. In current urban planning, there is insufficient consideration of the environment and public health. Therefore, in the future, exploring the impact of spatial correlation within cities or a region will be important with regard to the regional environment and public health, and governments should consider such factors to ensure rational allocation of urbanization resources.

## 5. Conclusions

Our study investigated the characteristics of urban spatial association and their evolution over time. It also revealed the relationships and mechanisms of urbanization among cities and provided an effective reference regarding the spatial development of urbanization in the YRD region. We integrated economic, population, and land urbanization factors to examine the spatial correlation characteristics of urbanization and its evolution. The YRD region had a clear distribution of city tiers, with Shanghai in the first tier (0.147–0.124), Nanjing, Hangzhou, and Suzhou in the second tier (0.045–0.059), and the remaining cities in the third, fourth, and fifth tiers. In addition, the strongest spatial correlation was concentrated in eastern parts of the YRD region, and 10 cities including Shanghai, Suzhou, Wuxi, and Hangzhou constituted the densest part of the urbanization spatial connection network.

Urbanization in the YRD region has been a dynamic process encompassing polarization enhancement and reduction. Overall, during the study period, it moved toward a more balanced distribution of development. The spatial evolution of urbanization in the YRD was characterized by the coexistence of cycles of polarization and diffusion. Analysis of the driving mechanisms revealed that the spatial characteristics of urbanization and its evolution in the YRD region were influenced by regional industrial modernization and relocation. Industrial modernization shaped the spatial hierarchy and differentiation of industries and underpinned the spatial diffusion dynamics of industries within the region.

China is in the process of rapid urbanization, and its spatial characteristics and evolutionary laws are influenced by the internal mechanisms of regional development. It is necessary to comply with the laws of geography in combination with the development stages of cities when formulating regional development policy. This might require appropriate coordination of regional urbanization, but we should not pursue balanced development of all cities within a region at the same time. In China, regional variation in urbanization is obvious, and additional data should be collected to allow further exploration of such differences.

## Figures and Tables

**Figure 1 ijerph-17-07276-f001:**
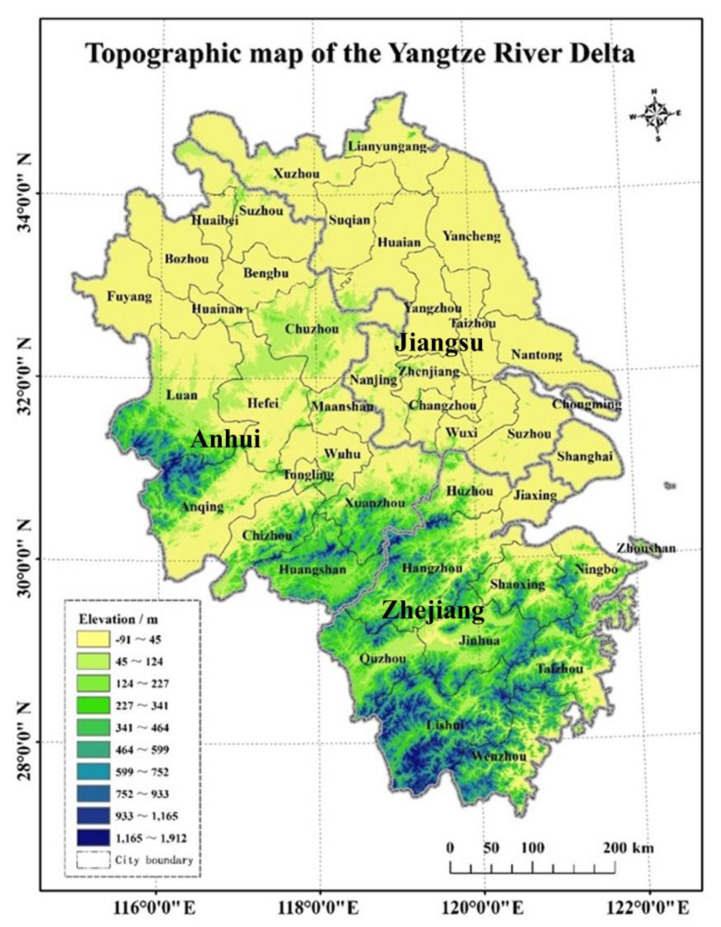
Topographic map of the YRD region.

**Figure 2 ijerph-17-07276-f002:**
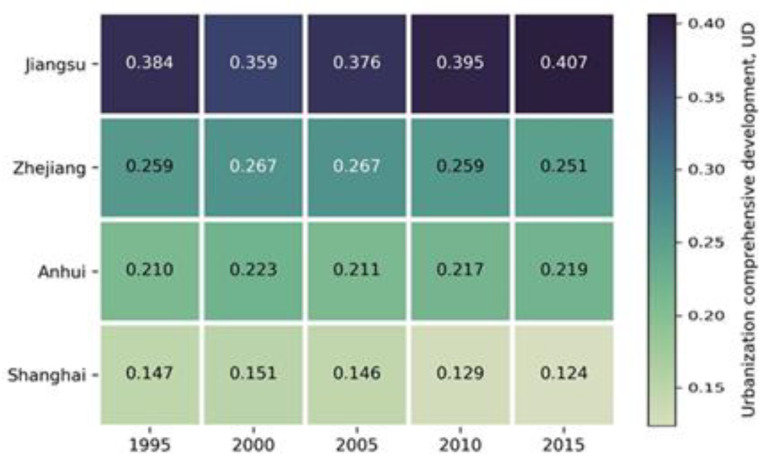
Overall urbanization development (UD) values of major cities in the YRD region.

**Figure 3 ijerph-17-07276-f003:**
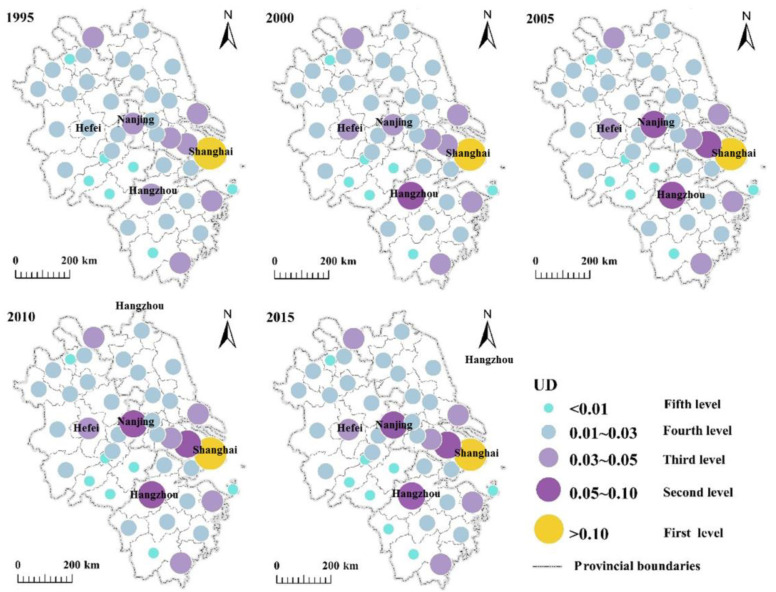
City-level structural maps of the YRD region.

**Figure 4 ijerph-17-07276-f004:**
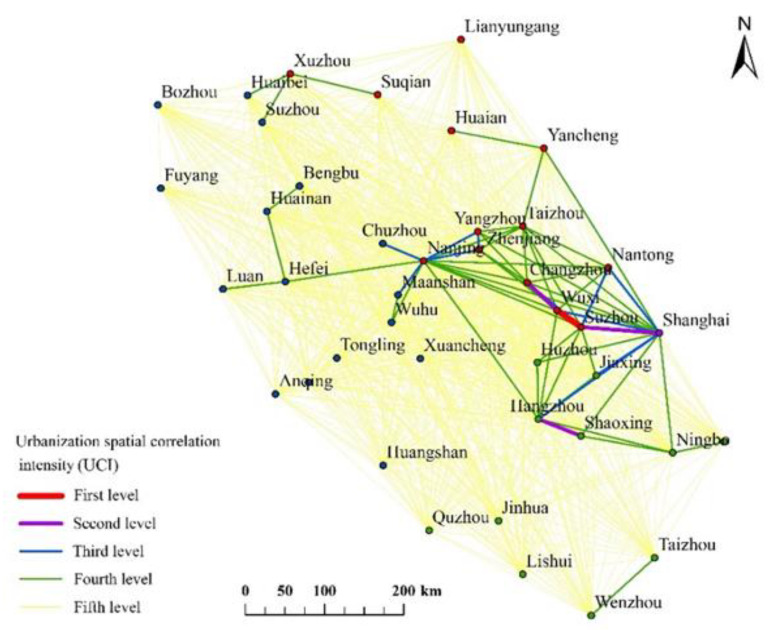
Spatial urban correlation intensity (UCI) map of urbanization in the YRD region.

**Figure 5 ijerph-17-07276-f005:**
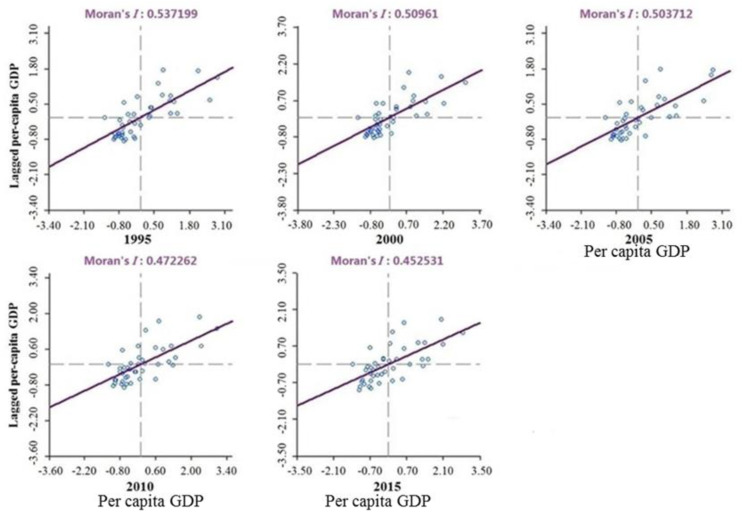
Moran’s *I* scatter plots in the YRD region from 1995 to 2015.

**Figure 6 ijerph-17-07276-f006:**
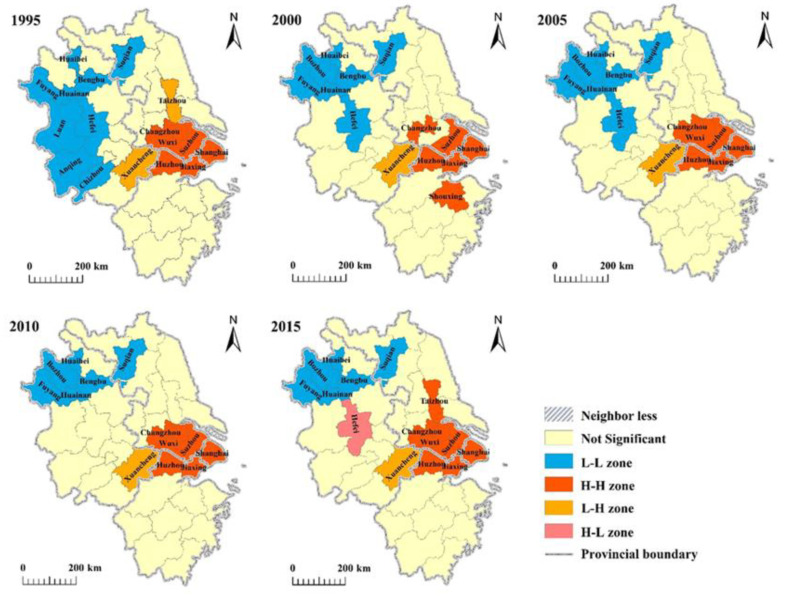
LISA agglomeration maps of the YRD region from 1995 to 2015.

**Figure 7 ijerph-17-07276-f007:**
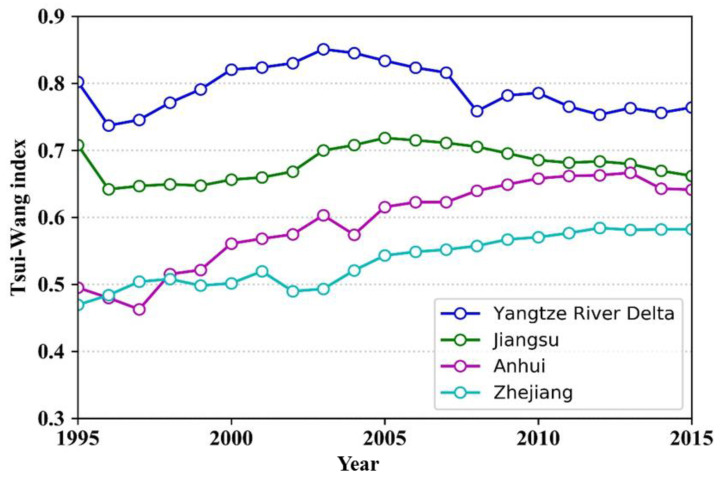
Comparison of the Tsui–Wang index of each province in the YRD region from 1995 to 2015.

**Figure 8 ijerph-17-07276-f008:**
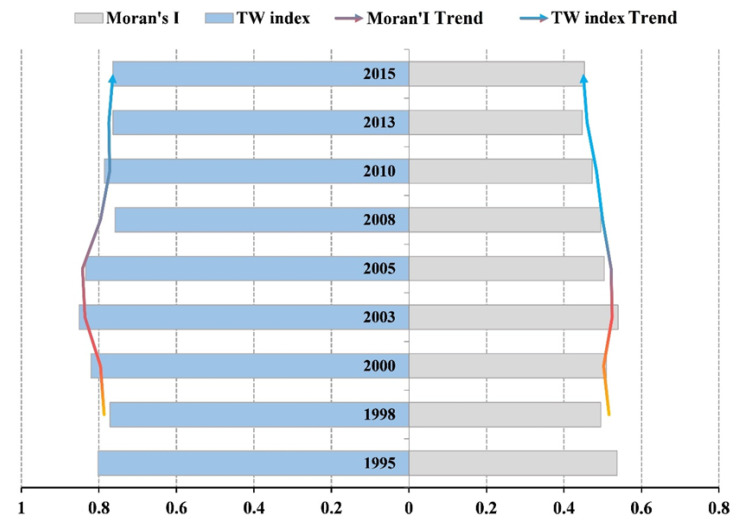
The Tsui–Wang index (TW) and Moran’s *I* in the YRD region (1995–2015).

**Figure 9 ijerph-17-07276-f009:**
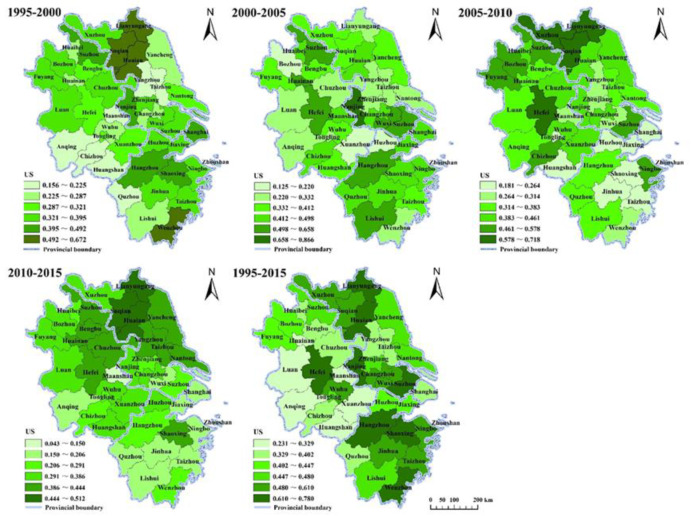
Urbanization speed (US) values of the YRD region from 1995 to 2015.

**Figure 10 ijerph-17-07276-f010:**
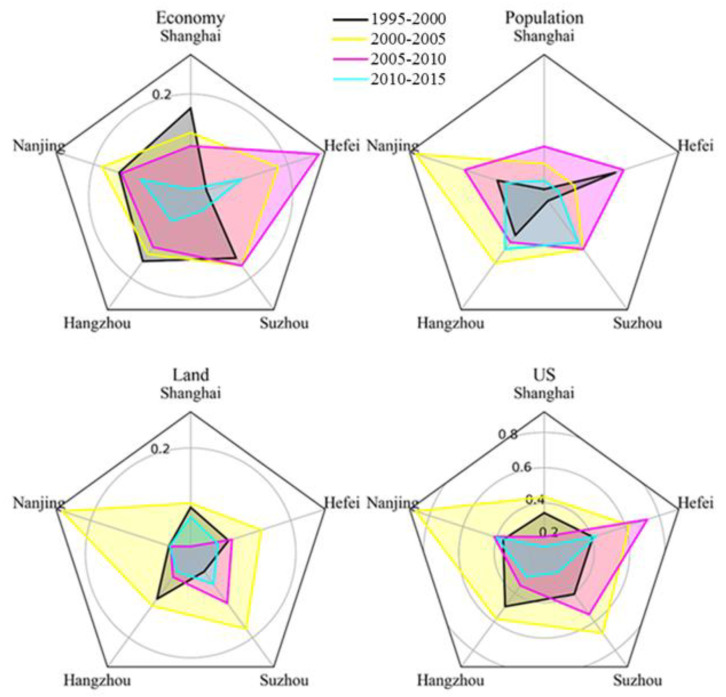
Radar map of economic, population, built-up area growth rate, and urbanization speed (US) values of the five major cities in the YRD region.

**Table 1 ijerph-17-07276-t001:** Overall urbanization development (UD) values of major cities in the YRD region.

Main Cities *	UD (1995)	UD (2000)	UD (2005)	UD (2010)	UD (2015)
Shanghai	0.147	0.151	0.146	0.129	0.124
Nanjing	0.049	0.049	0.061	0.059	0.059
Suzhou	0.047	0.043	0.053	0.058	0.058
Hangzhou	0.045	0.054	0.057	0.056	0.054
Hefei	0.029	0.033	0.036	0.041	0.042
Ningbo	0.037	0.036	0.037	0.043	0.040
Xuzhou	0.040	0.036	0.034	0.041	0.039
Wuxi	0.040	0.038	0.042	0.040	0.039
Nantong	0.038	0.033	0.031	0.032	0.035
Wenzhou	0.038	0.041	0.037	0.034	0.034

Note: * The main cities are those where the UD was always in the top 10 during the study period.

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
