# Peer review of "Spatial Association of Urbanization in the Yangtze River Delta, China"

_ijerph, 2020, doi:10.3390/ijerph17197276_

Round 1

Reviewer 1 Report

The manuscript is about assessing the spatial association of urbanization, economy, population and build-up growth in the Yangtze River Delta. The article is well developed and presented, but I recommend some adjustments to the text and figures.

Introduction: The authors address the phenomenon of spatial polarization in some countries and that there are regional differences in the provincial scale of China's urbanization and economy. However, the authors do not address the reason they chose to study the development of the Yangtze River Delta region. I recommend that you contextualize and describe the importance of the region studied in the introduction. In addition, I recommend that the authors highlight the objective of the study.

Figure 1. - I could not find Jiangsu, Zhejiang, Anhui in Figure 1.

Item 2.2. - Authors should describe only the data used. The second paragraph should be incorporated into another item on data analysis, perhaps in item 2.3.2.

Line 192 - The authors claim that urbanization development (UD) has not changed significantly but has not applied any statistical tests to affirm "significantly".

Figure 3. - Increase the resolution, as the map font is unreadable.

Conclusion: I recommend that the authors define an explicit objective to complete it. The conclusion of the manuscript resembles a summary of the results.

Reviewer 2 Report

By using spatial measurement related models, this study examined the spatial development process of urbanization in the Yangtze River Delta region from 1995 to 2015. On this basis, this study also explored the driving mechanism of regional urbanization. The whole research has certain value and contribution. The manuscript is within the journal scope. Only a couple of suggestions are acknowledged for its improvement:

1. The author needs to rewrite the abstract, which is too subjective and one-sided in evaluating existing research..

2. The author needs to reselect the key words, "City priority" and "regional radiation" should not be used as keywords, and should be replaced with "urbanization" and "inner mechanism".

2. The language of the manuscript must be improved. I recommend that the authors invite a native English speaker to edit the language.

3. The literature review is too weak.The authors need to introduce the existing research on urbanization spatial connection and its mechanism. And explain the reasons for the methods chosen in this paper.It is recommended that the author add more international literatures to improve this part.

4. L.64 The author believes that there is a significant difference between the level of urbanization and the level of economic development in China at the provincial level.Indeed, the urbanization differences between Chinese regions are very obvious, especially in the east and west. I suggest that the authors explore regional differences from all provinces in China, which is more valuable. The Yangtze River Delta region selected by the authors is the region with the most balanced urbanization development in China and it is not suitable as a case study to study regional urbanization differences.

5. In section 2.1, the authors need to further explain the representativeness of the Yangtze River Delta as a study area.

6. Since the units and attributes of each index are different, the data in this paper cannot be directly calculated, and the data needs to be standardized. The authors need to introduce the methods for standardizing data.

7. In the results section, the authors need to add the analysis of the reasons for the differences in the calculation results of different cities. At present, the authors just simply listed the results.

8. The conclusion part needs to be further improved. In addition to the need to summarize and analyze the results of this study, it is necessary to make a certain outlook on the future research direction.

The paper addresses an important problem. However, the presentation must be improved. I recommend major revision.

Reviewer 3 Report

The paper is clearly structured. The research results are presented in a transparent way.

However, I have two critical comments:

(1) Please expand your discussion/conclusion section and explain the potential impacts of your findings on environment and public health! Your study is about regional urban dynamics. However, you submitted your paper to the International Journal of Environmental Research and Public Health. This thematic link to Environment and Health is only mentioned in the first and last sentences of the introduction. That is not enough.

(2) Please interlink your results with the findings of Hu, Song, Li, and Zhang (2019) on "The Evolution of Industrial Agglomerations and
Specialization in the Yangtze River Delta from 1990–2018".  You claim that your findings depend on "industrial modernization and relocation" in the Yangzte River Delta -  this is exactly what Hu et al. were investigating.

Round 2

Reviewer 2 Report

The authors made some modifications to the article, but there are still some problems that can be further improved.

  1. The authors did not supplement the literature review section. There are many research publications on the spatial relevance and internal mechanism of urbanization, which are the basis of the research carried out in this article. In addition, through combing the existing methods, we can see the reasons why the authors used a specific method in this article.
  2. In the part of research methods, the authors did not explain clearly the reasons for selecting methods and indicators in related analysis in this article.
  3. In the results section, the authors graded the relevant results. What is the basis for doing so? What is the standard of division?
  4. The title of the article involves the internal mechanism of urbanization, but in the main text, the authors did not carry out much analysis and discussion on this, nor did they carry out discussions based on the analysis of urbanization autocorrelation.
